# Value of Glycemic Indices for Delayed Cerebral Ischemia after Aneurysmal Subarachnoid Hemorrhage: A Retrospective Single-Center Study

**DOI:** 10.3390/brainsci14090849

**Published:** 2024-08-23

**Authors:** Matthias Manfred Deininger, Miriam Weiss, Stephanie Wied, Alexandra Schlycht, Nico Haehn, Gernot Marx, Anke Hoellig, Gerrit Alexander Schubert, Thomas Breuer

**Affiliations:** 1Department of Intensive and Intermediate Care, Medical Faculty, RWTH Aachen University, 52074 Aachen, Germany; aschlycht@ukaachen.de (A.S.); nhaehn@ukaachen.de (N.H.); gmarx@ukaachen.de (G.M.); tbreuer@ukaachen.de (T.B.); 2Department of Neurosurgery, Medical Faculty, RWTH Aachen University, 52074 Aachen, Germany; miriam_weiss@icloud.com (M.W.); ahoellig@ukaachen.de (A.H.); gerrit.schubert@ksa.ch (G.A.S.); 3Department of Neurosurgery, Cantonal Hospital Aarau, 5001 Aarau, Switzerland; 4Institute of Medical Statistics, RWTH Aachen University, 52074 Aachen, Germany

**Keywords:** aneurysmal subarachnoid hemorrhage, delayed cerebral ischemia, blood glucose, dysglycemia, hyperglycemia, hypoglycemia, TWAG, ICU, TUDR, DCI

## Abstract

Delayed cerebral ischemia (DCI) is a severe complication following aneurysmal subarachnoid hemorrhage (aSAH), linked to poor functional outcomes and prolonged intensive care unit (ICU) stays. Timely DCI diagnosis is crucial but remains challenging. Dysregulated blood glucose, commonly observed after aSAH, may impair the constant glucose supply that is vital for brain function, potentially contributing to DCI. This study aimed to assess whether glucose indices could help identify at-risk patients and improve DCI detection. This retrospective, single-center observational study examined 151 aSAH patients between 2016 and 2019. Additionally, 70 of these (46.4%) developed DCI and 81 did not (no-DCI). To determine the value of glycemic indices for DCI, they were analyzed separately in patients in the period before (pre-DCI) and after DCI (post-DCI). The time-weighted average glucose (TWAG, *p* = 0.024), mean blood glucose (*p* = 0.033), and novel time-unified dysglycemic rate (TUDR140, calculated as the ratio of dysglycemic to total periods within a glucose target range of 70–140 mg/dL, *p* = 0.042), showed significantly higher values in the pre-DCI period of the DCI group than in the no-DCI group. In the time-series analysis, significant increases in TWAG and TUDR140 were observed at the DCI onset. In conclusion, DCI patients showed elevated blood glucose levels before and a further increase at the DCI onset. Prospective studies are needed to confirm these findings, as this retrospective, single-center study cannot completely exclude confounders and limitations. In the future blood glucose indices might become valuable parameters in multiparametric models to identify patients at risk and detect DCI onset earlier.

## 1. Introduction

Aneurysmal subarachnoid hemorrhage (aSAH) is a life-threatening stroke form that often requires prolonged treatment in the intensive care unit (ICU) [1]. About one-third of patients experience delayed cerebral ischemia (DCI), which may be associated with poorer neurological outcomes, longer ICU stays, and increased mortality [2,3,4]. DCI typically occurs between day three and fourteen after an aSAH event and can manifest as a reduction in vigilance or a new focal neurological deficit, often based on cerebral perfusion deficits [5,6]. DCI is hypothesized to result from multiple interrelated mechanisms, such as macrovascular and microvascular dysfunction, causing vasospasm, microthrombosis, neuroinflammation, and cortical spreading depolarization [6,7]. Various procedures are used to diagnose DCI as soon as possible [8]. In addition to transcranial Doppler sonography [9] and electroencephalography [10], invasive procedures such as cerebral microdialysis, cerebral oxygen monitoring [11], and computed tomography (CT) perfusion imaging [12] are utilized. Identifying patients at increased risk is vital for achieving the targeted use of these procedures. 

The modified Fisher score (mFisher score), which is widely used, indicates a more severe outcome. Nevertheless, it suffers from interrater variability and is calculated only once upon admission [13]. Therefore, it is valuable to identify potential complementary objective prognostic variables for DCI. To date, no single objective continuous parameter has surpassed the established mFisher score in DCI risk prediction; therefore, multiparameter models might be key [14]. 

Blood glucose is vital in this context as the brain depends on its constant supply [15]. Critical illnesses, including aSAH, often involve glucose dysregulation, most notably stress-induced hyperglycemia [16]. Blood glucose levels serve as treatment targets and potential prognostic markers. Numerous studies have investigated the role of blood glucose in predicting clinical outcomes after aSAH [17]. 

Evidence that hyperglycemic values above 216 mg/dL might predict DCI was first found in 1999. However, the temporal context of DCI was not adequately considered [18]. Subsequent studies used hyperglycemic cutoffs of 125–140 mg/dL to categorize patients dichotomously as normo- or hyperglycemic, based on either admission blood glucose [19,20,21,22,23], two to three fasting blood glucose values within the first 7 days [20], the mean blood glucose level over the first 14 days [21], or the entire hospital stay [23,24]. In line with the heterogeneity of the cutoffs and included values, the results varied, as some studies could prove a correlation with the DCI [20,21,22], while others could not [19,24]. Two studies found significantly higher maximum glucose values in DCI patients within the first 24 h [25] or over the entire stay [26]. For minimum blood glucose values under 80 to 90 mg/dL, an independent association with DCI was shown over the entire hospital stay [23,27]. To date, however, no association between glucose variability and DCI, but with mortality, has been demonstrated in aSAH patients when including the entire hospital stay [28,29,30].

These studies had several methodological weaknesses as follows: average blood glucose over the entire stay or a fixed number of days lacked the ability to draw conclusions about whether hyper- or hypoglycemia occurred before or after the DCI event. The use of the first recorded, maximal, or minimal value is also subject to numerous biases, including selection bias, and therefore does not provide a valid basis for analysis. Moreover, most studies categorized patients dichotomously as hyperglycemic using cutoff values, resulting in an inadequate capture of the dynamic blood glucose curve. None of the studies considered hyperglycemia and hypoglycemia simultaneously. A time-differentiated rather than solely dichotomous analysis of the data, accounting for blood glucose target corridors, seems desirable to determine whether blood glucose levels leading to DCI can be helpful in predicting or detecting it.

Therefore, our study aimed to investigate whether and, if so, which blood glucose indices in aSAH patients can help identify patients at risk of DCI. Additionally, we analyzed dysglycemic rates, fluctuations in blood glucose levels, and blood glucose trends over time in relation to DCI onset.

## 2. Materials and Methods

### 2.1. Study Design and Patient Selection

Approved by the Ethics Committee of the Medical Faculty, RWTH Aachen University (approval date: 28 July 2020, approval number: EK275/20), the study waived the requirement for written informed consent as retrospective data were analyzed entirely anonymously. All research procedures were conducted in accordance with institutional ethical standards, relevant guidelines and regulations, and the Declaration of Helsinki. This article was created according to the Strengthening the Reporting of Observational Studies in Epidemiology (STROBE) guidelines [31].

The endpoint of this study was the development of DCI. All patients admitted to the neurosurgical ICU at the Department of Intensive and Intermediate Care, RWTH Aachen University between 2016 and 2019 were screened for SAH. To analyze blood glucose in relation to DCI onset, this study included aSAH patients admitted to the hospital within 72 h after the event. Exclusion criteria included patients without aSAH, those with DCI occurrence after the observation period (>14 days), and patients who died or were considered moribund and thus were not treated to the full extent within the first 72 h after ICU admission.

### 2.2. Data Collection

Baseline data for all 151 patients were extracted from medical files using a Microsoft Excel spreadsheet (Microsoft 365 MSO, Version 2112, Redmond, WA, USA). Variables were collected as follows: (a) demographic data, including age, sex, and body mass index; (b) medical history: diabetes, arterial hypertension, psychiatric disorder, stroke history, history of SAH, chronic kidney failure, chronic liver disease, alcohol addiction, and smoking history; (c) details of aSAH: Hunt and Hess score, mFisher score, aneurysm localization, and endovascular or surgical treatment; and (d) details of ICU stay: simplified acute physiology score (SAPS II) on ICU admission, mechanical ventilation hours, ICU and hospital length of stay, and ICU mortality.

### 2.3. Standard Treatment Algorithm

Our aSAH treatment protocol has been published previously [11,32]. DCI was diagnosed using two entities, both of which may establish a DCI diagnosis [11]. In brief, in awake patients, DCI was defined after the criteria set by Vergouwen et al. as a reduction in vigilance with a decrease of at least two points on the Glasgow Coma Scale for at least one hour or a new focal neurological deficit [5]. In comatose patients, DCI was diagnosed equally if a CT perfusion deficit with territorial or watershed hypoperfusion was apparent (time to drain >10 s, mean transit time >6.7 s) [33]. DCI treatment included induced hypertension (systolic blood pressure target >180 mmHg) and, in refractory cases, endovascular rescue treatment with continuous intraarterial nimodipine and/or balloon angioplasty [32]. 

### 2.4. Determination of the Times for aSAH Event and DCI Onset

Most included patients were part of a prospective observational study (*n* = 109/151, 72.2%, with DCI *n* = 51/70, 72.9%) [32,33]. To ensure the quality of the temporal analyses, hemorrhage and DCI time points were acquired from this database, where they were documented prospectively by a neurosurgeon. The time points of the remaining patients were reconstructed from their medical records. The time point of DCI was generally considered the time point of clinical deterioration or the time stamp of CT perfusion.

### 2.5. Analysis of Blood Glucose Values

Glucose values were included only from blood gas analysis (ABL90 FLEX, Radiometer, Krefeld, Germany) in our neurosurgical ICU due to the higher frequency and exact time stamp of measurements compared to laboratory sampling to ensure measurement uniformity. Measurements were recorded automatically using an electronic patient data management system (IntelliSpace Critical Care and Anesthesia, Koninklijke Philips N.V., Amsterdam, The Netherlands). All blood glucose values from ICU admission to the end of the assumed DCI window (maximum of 14 days) were collected. Datasets were analyzed for the total monitoring period in all patients and were additionally split for each DCI patient into periods before (pre-DCI) and after DCI (post-DCI) onset.

The following glycemic indices were calculated:(1)Mean blood glucose (MBG) [34];(2)Time-weighted average glucose (TWAG) [34,35,36];(3)Coefficient of glycemic variation (CV) [37];(4)Average absolute change by time difference (AACTD) [29];(5)Time-unified dysglycemic rate (TUDR).

These glycemic indices were selected to effectively describe, analyze, and compare blood glucose levels from a retrospective cohort lacking standardized, uniform blood glucose measurement time points. Besides the well-described (1) MBG calculated by averaging all glucose measurements per patient [34], additional indices were included, as different patients might have frequent measurement repetitions or varying measurement intervals, which could influence the MBG and bias side-by-side comparisons of these patients. To address this limitation, (2) the TWAG was calculated for every patient. In addition to the blood glucose level, this index considers the time between measurements. The TWAG is calculated as the area under the blood glucose curve divided by the number of hours analyzed [34,35,36]. This enabled comparability between patients with different lengths of ICU stay. In addition to absolute blood glucose levels, it is worth considering the variation between consecutive blood glucose measurements within a single patient, the so-called fluctuation. Hence, the (3) CV was calculated as the glucose standard deviation divided by the MBG [37] and the (4) AACTD was analyzed. The AACTD considers the level of blood glucose fluctuation as the time between measurements, calculated as the average absolute change in blood glucose divided by the inter-measurement time [29]. 

Finally, the novel (5) TUDR was developed and applied in this study for the first time. The TUDR aimed to determine the time periods outside the specified blood glucose range relative to the total monitoring time. This allowed a more in-depth analysis of blood glucose over time and the simultaneous consideration of hyper- and hypoglycemia for individual patients or in relation to events like the onset of DCI across all patients. Unlike comparing absolute times outside the range to those within, the TUDR uses fixed time periods, which mitigate the negative impact of unfavorable timing and low-frequency measurements that could distort the glucose profile when analyzing absolute time differences. Thus, fixed time periods, as used in the TUDR, provide a more robust and reliable assessment, improving comparability within and between patients (Figure 1). 

Before calculating the TUDR, the data of every patient were divided into periods of 1, 3, 6, and 12 h. The optimal period was determined by calculating the percentage of patients with at least one blood glucose measurement per period using a 95% median coverage threshold and selecting the shortest period. A period was considered dysglycemic if any blood glucose value was outside the target range (hypoglycemic [<70 mg/dL] or hyperglycemic [>140/160/180 mg/dL]). 

The TUDR was calculated as follows: (I) as individual TUDRs by calculating the ratio of dysglycemic to total periods for each patient starting from ICU admission and (II) as cumulative TUDRs per period by the proportion of patients with at least one blood glucose value outside the target range in the respective period by the total number of patients in this period starting from the onset of DCI. 

### 2.6. Statistics

Statistical analyses were performed using SPSS (version 28, IBM Corp., Armonk, NY, USA). Glycemic indices were calculated using R (version 4.2.0) and RStudio (version 2024.04.2 Build 764, Boston, MA, USA). To examine differences in baseline patient characteristics between the DCI and no-DCI groups, unpaired t-tests were performed for continuous parameters. For BMI, a bootstrapped *t*-test was performed due to skewed data. For blood glucose indices, the Mann–Whitney U test was used to compare groups, and the Wilcoxon matched-pairs signed-rank test to compare pre and post periods within the DCI group. Differences in the frequencies of binary variables between the DCI and no-DCI group were evaluated using Fisher’s exact test. To assess the prediction of DCI using glucose indices, a multivariable regression analysis was performed. Therefore, the correlation was first assessed using Pearson’s correlation coefficient. Second, a univariate logistic regression of the glucose indices (TWAG, AACTD), as well as the mFisher score, Hunt and Hess score, age, and sex was conducted, except for the parameters with a high correlation (r ≥ 0.50). All predictors with a *p*-value < 0.050 were considered candidates for multivariable regression. Interrupted time-series analysis was performed to investigate the role of blood glucose in the timely context of DCI onset. When appropriate, data are shown as mean ± SD, median (interquartile range), or absolute numbers and frequencies. All statistical tests were two tailed. Although the present analysis was exploratory, a *p*-value < 0.050 was considered statistically significant. Figure 1 was created using BioRender.com. The graphical presentation of the data was performed using Prism 10.2.3 (GraphPad Software Inc., San Diego, CA, USA).

## 3. Results

### 3.1. Study Population

The medical charts of 462 patients admitted to the neurosurgical ICU at the Department of Intensive and Intermediate Care, RWTH Aachen University, Germany, between 2016 and 2019, with a diagnosis of SAH according to the International Classification of Diseases (ICD60), were screened (Figure 2). 151 patients met the inclusion criteria and were included in this study. Of these, 70 patients (46.4%) had DCI (DCI group) and 81 did not (no-DCI group).

Patient characteristics, including medical history, SAH scores, treatment, aneurysm location, and ICU key facts, are shown in Table 1. Patients in the DCI group were significantly younger than patients without DCI (DCI: 52.9 ± 11.8 years; no-DCI: 57.2 ± 12.6 years; *p* = 0.035) and had an increased mFisher score (DCI: 2.9 ± 1.1; no-DCI: 2.2 ± 1.3; *p* < 0.001).

Moreover, a significantly longer ICU (DCI: 31.7 ± 16.5 days; no-DCI: 19.6 ± 10.8 days; *p* < 0.001) and hospital stay (DCI: 36.1 ± 18.3 days; no-DCI: 25.2 ± 12.1 days; *p* < 0.001), higher ICU mortality (DCI: *n* = 19 (27.1%); no-DCI: *n* = 9 (11.1%); *p* = 0.020), higher SAPS II on admission (DCI: 39.1 ± 9.2; no DCI: 36.0 ± 9.0; *p* = 0.039), and longer mandatory ventilator time (DCI: 517.6 ± 459.3 h; no-DCI: 235.8 ± 328.0 h; *p* < 0.001) could be shown for the DCI group.

### 3.2. Increased Blood Glucose Values before DCI Onset

MBG was significantly lower in the no-DCI group when compared to the different time periods of the DCI group (no-DCI: 125.6 (112.5–137.0) mg/dL; pre-DCI: 130.7 (122.1–143.8) mg/dL, *p* = 0.033; post-DCI: 149.2 (137.3–161.4) mg/dL, *p* < 0.001; total time DCI: 140.3 (128.5–152.9) mg/dL, *p* < 0.001). Moreover, MBG levels were significantly lower in the pre-DCI than in the post-DCI period of the DCI group (Figure 3a, *p* < 0.001).

The analysis of the TWAG showed significant results comparable to the MBG with slightly lower values (no-DCI: 124.3 (110.7–136.4) mg/dL; pre-DCI: 128.7 (121.2–144.6) mg/dL, *p* = 0.024; post-DCI: 149.4 (136.9–160.0) mg/dL, *p* < 0.001; total time DCI: 138.1 (128.6–152.6) mg/dL, *p* < 0.001, Figure 3b). Furthermore, the TWAG of the pre- and post-DCI periods was significantly different, with higher levels post-DCI (*p* < 0.001).

### 3.3. No Difference in Blood Glucose Fluctuation before DCI Onset

The glycemic variability indicated by the CV did not significantly differ between patients without DCI and the pre- and post-DCI periods in the DCI group (Figure 4a), nor between the pre- and post-DCI periods. However, the CV was significantly higher in the DCI group during the total monitoring period compared to the no-DCI group (no-DCI: 16.9 (13.6–19.5)%, total time DCI: 18.6 (16.6–21.8)%, *p* < 0.001). The AACTD showed similar results when comparing the no-DCI and pre-DCI periods (Figure 4b). Significant differences were found between the no-DCI group and the post- and total time period of the DCI group, as well as between the pre- and post-DCI periods (no-DCI: 5.2 (3.9–6.4) mg/dL/h, pre-DCI: 5.5 (4.5–7.2) mg/dL/h, *p* = 0.135; post-DCI: 7.6 (5.5–11.1) mg/dL/h, *p* < 0.001; total time DCI: 7.0 (5.7–9.0) mg/dL/h, *p* < 0.001; for pre-DCI versus post-DCI period, *p* < 0.001).

### 3.4. Higher Time-Unified Dysglycemic Rate before DCI Onset

Given the significantly higher summative glucose indices in the pre-DCI group compared to the group without DCI, a more detailed temporal analysis was subsequently conducted. In addition to hyperglycemia, hypoglycemia was also considered by setting a blood glucose target range. Due to the differences in the average blood glucose measurement intervals between the no-DCI and DCI group (no-DCI: 4.0 ± 1.6 h, DCI: 2.8 ± 0.8 h, *p* < 0.001), neither absolute nor relative quantifications of dysglycemic blood glucose were directly comparable without the risk of inaccuracy. Hence, the TUDR was established, as it divides ICU stays into equal time periods to ensure comparability. It summarizes the blood glucose curve as a single value by calculating the ratio of dysglycemic to all periods.

Six hours was determined to be the optimal time period length for calculating the TUDR (1 h: 29.8%, 3 h: 77.5%, 6 h: 98.0%, 12 h: 100.0%). With a target blood glucose range of 70–140 mg/dL (TUDR140), the no-DCI group had a significantly lower median time rate spent in dysglycemia compared to the pre-DCI period of the DCI group (no-DCI: 24.1 (11.3–50.7)%; pre-DCI: 35.8 (15.5–65.5)%, *p* = 0.042; Figure 5a). For the target range of 70–160 mg/dL (TUDR160, no-DCI: 7.0 (1.9–17.1)%, pre-DCI: 12.3 (3.2–29.5)%, *p* = 0.068; Figure 5b) and 70–180 mg/dL (TUDR180, no-DCI: 2.0 (0.0–6.8)%, pre-DCI: 5.6 (0.0–12.6)%, *p* = 0.190; Figure 5c), no significant group difference was shown before DCI onset. Compared with the no-DCI group, a significantly higher rate of dysglycemic periods was observed with all three cutoffs for the DCI group when considering the post-DCI period or the total time.

### 3.5. Time-Weighted Average Glucose Associated Positively with DCI

Using logistic regression, possible predictors of DCI were examined. Age and SAPS II score (r = 0.50), CV and AACT (r = 0.53), as well as MBG, TWAG (r = 0.99), and TUDR140 (r = 0.92) were positively correlated; therefore, only the age, AACT, and TWAG were used in the regression analysis due to their lower *p*-values. Of all the parameters studied (Appendix A), only the TWAG (OR = 1.020, 95%-CI [1.002, 1.039], *p* = 0.029), mFisher score (OR = 3.967, 95%-CI [2.006, 7.843], *p* < 0.001), and age (OR = 0.972, 95%-CI [0.946, 0.998], *p* = 0.039) were significantly associated in univariate logistic regression. In the multivariable logistic regression, an increase in the TWAG was positively associated with DCI occurrence (OR = 1.019, 95%-CI [0.998–1.040], *p* = 0.073), but not statistically significant. Lower age (OR = 0.961, 95%-CI [0.932–0.990], *p* = 0.009) and higher mFisher scores (OR = 3.966, 95%-CI [1.930–8.152], *p* < 0.001) were significantly associated with DCI.

### 3.6. Time-Weighted Average Glucose and Time-Unified Dysglycemic Rate as Marker for DCI Onset

To analyze the blood glucose profile before and after the onset of DCI closer for all DCI patients, the cumulative TUDR and TWAG were calculated. DCI onset was defined as time point zero, and all patients with 6 h period values for the TWAG and TUDR up to 7 days (168 h) before (negative time) and after (positive time) DCI onset were plotted on a timeline. Both the TWAG (pre-DCI [−6 h]: 127.7 (114.9–147.3) mg/dL, post-DCI [6 h]: 136.0 (125.0–161.0) mg/dL, *p* = 0.017; Figure 6a) and the cumulated TUDR140 (pre-DCI [−6 h]: 41.4%, post-DCI [6 h]: 58.6%, *p* = 0.016; Figure 6b) showed a significant increase in the time period directly after DCI onset when compared to the last pre-DCI period, whereas the TUDR160 (pre-DCI [−6 h]: 20.0%, post-DCI [6 h]: 37.1%, *p* = 0.359) and TUDR180 (pre-DCI [−6 h]: 7.1%, post-DCI [6 h]: 20%, *p* = 0.260) did not. Interrupted time-series analysis showed a significant increase in the cumulative TWAG (*p* < 0.001), as well as all three blood glucose ranges (TUDR140: *p* = 0.002; TUDR160: *p* < 0.001; TUDR180: *p* < 0.001) at the DCI onset, but there was no significant trend within the periods before or after DCI.

The blood glucose increase at DCI onset was conclusively investigated by a qualitative view of all blood glucose measurements of DCI patients 12 h before and after DCI onset (Appendix A). In this timeframe per hour, 34.7 ± 4.7% of DCI patients had at least one blood glucose measurement. For both the cumulative TWAG and TUDR, a trend towards higher levels was observed five to seven hours after DCI onset. The statistical analysis of hourly glucose data was waived because of the limited number of measurements per period.

## 4. Discussion

This study was the first to evaluate the temporal association between blood glucose and DCI. Previous research neglected temporal relations, particularly around DCI onset, fluctuations in blood glucose, and used divergent hyperglycemic cutoffs with dichotomous group assignments [20,21,22,23,38]. We observed elevated blood glucose levels before and a further increase at DCI diagnosis compared to patients without DCI. 

The period before DCI onset is of particular importance, as identifying patients at risk and diagnosing DCI as early as possible are key challenges [8]. Thus, the significantly higher glycemic indices of the TWAG, MBG, and TUDR140 could be helpful in improving DCI risk stratification. Moreover, blood glucose indices may enhance early DCI detection as time-series analysis revealed a significant rise in blood glucose indices at DCI onset.

Pathological cascades that precede clinical events, like DCI, may be induced by prolonged dysglycemia, as indicated by the data of this study. In vivo studies in rats showed that hyperglycemia increases cerebral vasospasm, infarct volume, cerebral ischemia, blood–brain–barrier damage, and neutrophilic infiltration within the first 72 h after SAH [39,40]. Similar mechanisms may contribute to delayed brain injury like DCI. In addition, the further increase in glucose at DCI onset could be explained as part of a more pronounced stress response through increased sympathetic activation or increased cortisol levels [16,41].

In recent years, the number of potential DCI predictors and indices increased, including combinations of demographic, clinical, laboratory, as well as diagnostic inputs [42,43]. However, a recently published systematic review concluded that, despite some identified risk factors, their likelihood of bias is high, and no adequate predictive model for DCI could be identified [44]. This may be due to the multifactorial nature of DCI [6] and the difficulty to derive continuous measures from the brain directly, which necessitates the consideration of multiple factors for an accurate prediction. Thus, it seems plausible that recent studies highlighted the advantages of machine learning models that incorporate various risk factors and outperform conventional models [45,46]. If future research confirms the findings of our feasibility study, incorporating blood glucose indices as additional factors into these multimodal models seems reasonable.

The cutoff value for diagnosing hyperglycemia in aSAH patients varies widely (e.g., 99–200 mg/dL) across studies [38]. The European Stroke Organization guidelines recommend insulin treatment only if the blood glucose level exceeds 180 mg/dL [47]. Using three different hyperglycemic cutoffs, we could show significant differences in the rates of dysglycemia for 140 mg/dL as the cutoff, but not for 160 and 180 mg/dL, both when comparing patients pre-DCI and without DCI, as well as time periods directly before and after DCI onset in time-series analysis. Consistent with studies on aSAH mortality [27,48,49], our data indicated that a stricter hyperglycemic cutoff could improve disease prognostication in these patients. 

If prospective studies validate these findings, this might imply tight blood glucose control in the ICU for patients at risk for DCI. Bilotta and colleagues showed in a small, prospective monocentric study that tight blood glucose regulation (80–120 mg/dL) compared to moderate regulation (80–220 mg/dL) did not affect vasopasm incidence [50]. The study, however, faced significant limitations, including a threefold increase in hypoglycemia in the tight glucose control group. Therefore, it seems reasonable for future studies to (1) quantify actual blood glucose profiles, for example by using indices like the TUDR, and (2) sufficiently prevent dysglycemia. The latter, in particular, appears to be an essential prerequisite for successful tight blood glucose regulation. It can be achieved by using automated blood glucose measurement system to promptly detect out-of-range glucose. Ideally, this system should be blood saving and incorporate closed-loop blood glucose regulation to prevent dysglycemia, particularly hypoglycemia. Such systems are currently undergoing preclinical testing [51]. Hence, further investigation is needed to determine whether lower hyperglycemic cutoffs should be used as an indicator for therapeutic regulation and, if so, to identify the optimal cutoff for specific implications. 

We note that this retrospective study has several limitations. HbA1c is not routinely measured in our clinic; therefore, references for pre-SAH blood glucose history were absent and the glycemic gap could not be calculated. The number of patients with a known history of diabetes was, however, relatively low. Glucose sampling intervals were not standardized. This limitation was addressed by establishing indices such as the TWAG and AACTD and by unifying glycemic data over time using the TUDR. Drugs affecting blood glucose, nutrition, or insulin therapy were not considered. Additionally, our study showed a higher incidence of DCI (46.4%) than reported in the literature due to the inclusion of radiographic hypoperfusion in the DCI definition. The lack of a standardized DCI definition for comatose patients may have introduced detection bias and limits the comparability with other studies that used DCI as an endpoint [3]. The retrospective single-center design of this study may introduce environmental and geographical biases, as it analyzed patient data solely from a large university neurosurgical ICU in Germany, potentially limiting result transferability. Despite thorough data collection, not all potential confounders could be considered as the inherent heterogeneity of patients and their received treatment remains. Although optimal patient selection was intended, the inclusion criteria might have introduced selection bias. Consequently, the total number of patients included was relatively small in comparison to other acute illnesses requiring ICU treatment, which could impact the statistical power. 

Given that the study aimed to demonstrate the feasibility of time-differentiated blood glucose analysis in the context of DCI to generate initial hypotheses, its limitations can be resolved in future, more extensive multicenter studies. Therefore, larger patient groups should be observed prospectively, with time-stamped records of potential confounders affecting blood glucose or insulin sensitivity, including medication, nutrition, interventions, and complications, such as infections. In addition, standardized, frequent blood glucose monitoring should be implemented to enhance comparability. Continuous, non-blood-consuming glucose sensors could therefore be used in the future [51]. Finally, supplementing systemic glucose data with cerebral tissue glucose information, like cerebral microdialysis measurements, could provide insight into the relationship between glucose dynamics and DCI. 

## 5. Conclusions

This study demonstrated that hyperglycemia and dysglycemia were more prevalent before DCI, with a further increase afterward. Blood glucose indices may provide more informative value than the absolute blood glucose levels alone. Thus, blood glucose indices could potentially support risk stratification and DCI detection in clinical practice when combined with other known parameters, ideally within multiparametric risk models. However, prior to this, the findings of this single-center retrospective feasibility study require validation in a larger prospective cohort, excluding potential confounders.

## Figures and Tables

**Figure 1 brainsci-14-00849-f001:**
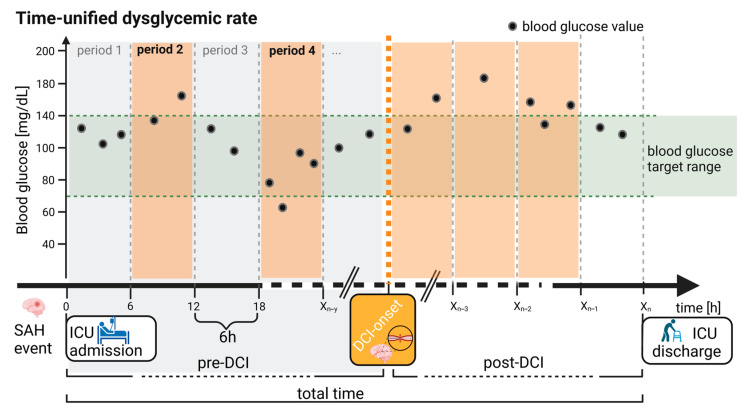
Exemplary illustration of time-unified dysglycemic rate for a DCI patient. An exemplary calculation of the time-unified dysglycemic rate (TUDR) for a period length of 6 h is depicted. The time in hours was plotted on the *x*-axis and the blood glucose level on the *y*-axis. Time periods are illustrated by vertical gray dashed lines. The target blood glucose range (here, 70–140 mg/dL) is shown as a horizontal green area. The DCI onset is represented by an orange vertical line, which divides the monitoring time (total time, X_n_) into time before (pre-DCI) and after DCI (post-DCI). Each blood glucose value is represented by a black dot on the graph. Dysglycemic periods, in which at least one measured value was outside the blood glucose target range, are highlighted in orange. The TUDR was calculated by dividing the number of dysglycemic by the total number of periods. For the pre-DCI time in this example, a TUDR of 40% resulted as two (periods 2 and 4) of the five periods were dysglycemic. ICU: Intensive care unit, SAH: Subarachnoid hemorrhage.

**Figure 2 brainsci-14-00849-f002:**
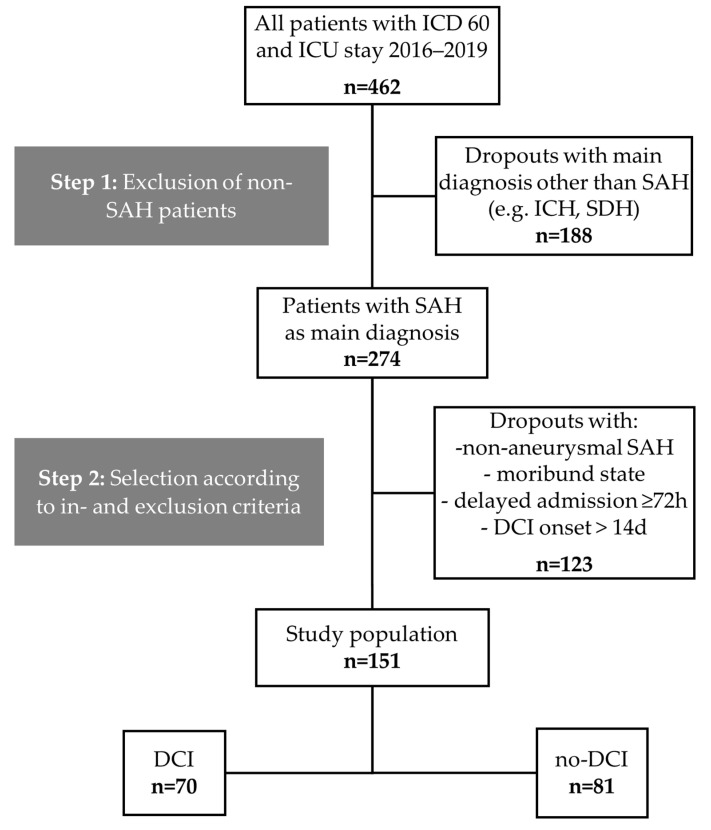
Study population flow chart. aSAH: Aneurysmal subarachnoid hemorrhage, DCI: Delayed cerebral ischemia, ICD: International Classification of Diseases, ICH: Intracranial hemorrhage, SDH: Subdural hematoma.

**Figure 3 brainsci-14-00849-f003:**
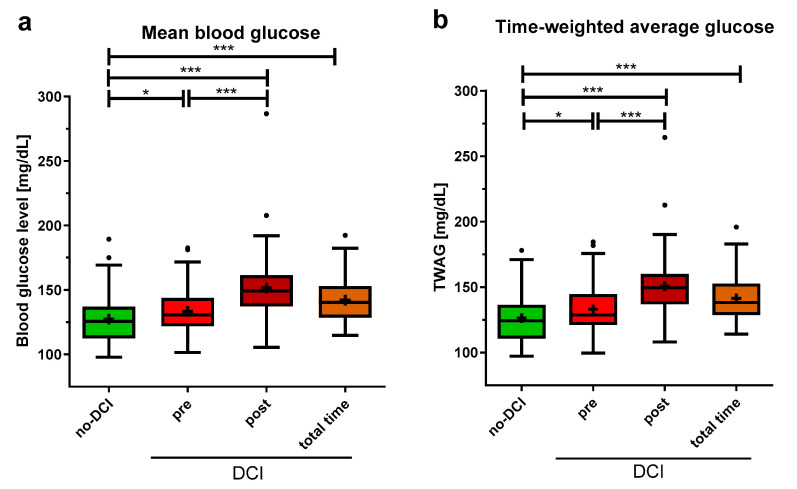
Average blood glucose indices in aSAH patients with and without DCI. (**a**) For patients without DCI (no-DCI), blood glucose was averaged over 14 days, for DCI patients, this was split up: period before DCI event (pre), after DCI event (post), and the total 14 observation days (total time). (**b**) Time-weighted average glucose (TWAG) was calculated as the area under the blood glucose curve. Data are presented as median ± 25th to 75th percentiles (box) with Tukey whiskers; data outside the Tukey range are shown as separate dots; mean is shown as +; * *p* < 0.050, *** *p* < 0.001.

**Figure 4 brainsci-14-00849-f004:**
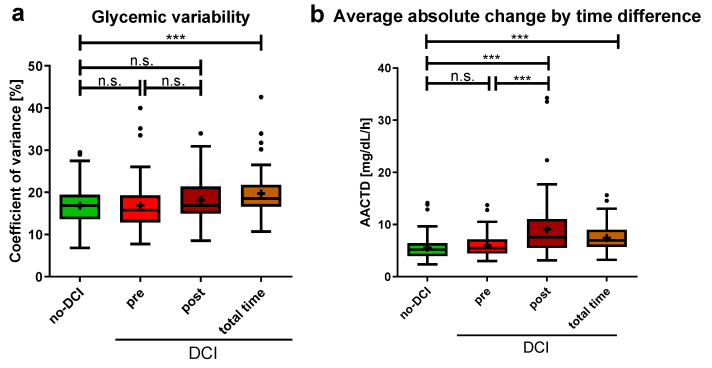
Blood glucose fluctuations in aSAH patients with and without DCI (**a**) Coefficient of glycemic variation is shown. For no-DCI patients, over 14 days, for DCI patients, split up: pre-DCI, post-DCI, and the total 14 monitoring days. (**b**) Average absolute change by time difference (AACTD) was calculated as the average absolute change in blood glucose divided by the inter-measurement time. Data are presented as median ± 25th to 75th percentiles (box) with Tukey whiskers, data outside the Tukey range are shown as separate dots; mean is shown as +; n.s., not significant; *** *p* < 0.001.

**Figure 5 brainsci-14-00849-f005:**
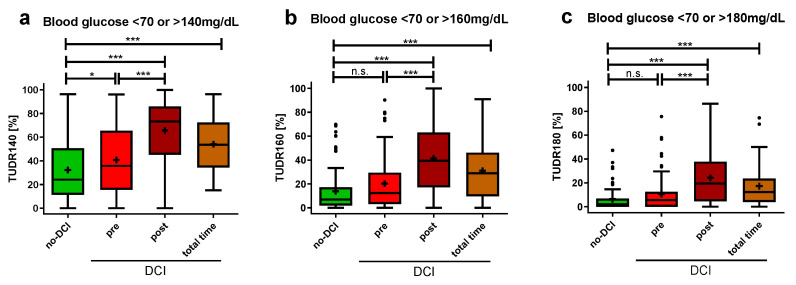
Time-unified dysglycemic rate in aSAH patients with and without DCI. Dysglycemic rates with 6-hourly periods are depicted with hyperglycemic cutoffs of (**a**) >140 mg/dL (**b**) >160 mg/dL (**c**) > 180 mg/dL and hypoglycemic cutoff of <70 mg/dL. All blood glucose values recorded in the ICU for up to 14 days were included. Glycemic values of DCI patients were divided into pre-DCI, post-DCI, and total time. The TUDR was calculated by dividing the dysglycemic by the total number of periods for each patient. The dysglycemic ratio is presented as median ± 25th to 75th percentiles (box) with Tukey whiskers, data outside the Tukey range are shown as separate dots; mean is shown as +; n.s., not significant; * *p* < 0.050, *** *p* < 0.001.

**Figure 6 brainsci-14-00849-f006:**
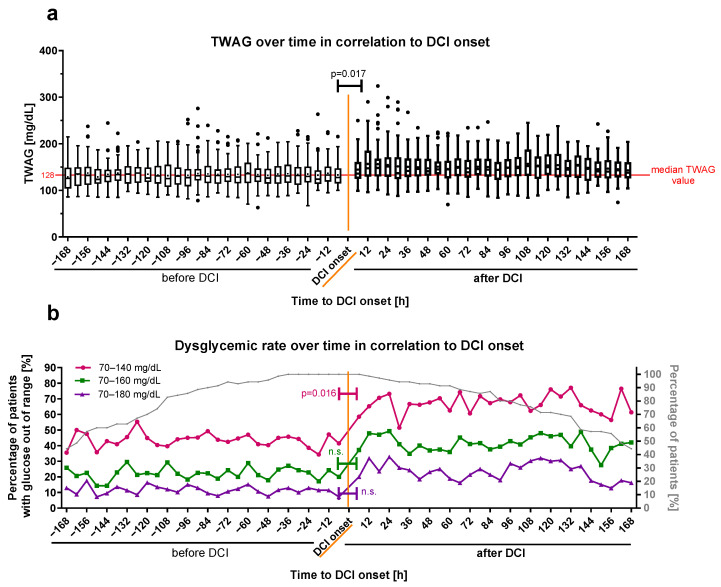
Glucose profile time-series of DCI patients before and after DCI onset. (**a**) The cumulated TWAG and (**b**) TUDR for 6-hourly periods of patients suffering from DCI are plotted in relation to its onset (time point zero in orange). Periods before DCI onset were defined as negative and those after the onset as positive. Dysglycemic rates are illustrated with cutoffs <70 mg/dL and >140 mg/dL (pink color), >160 mg/dL (green color), or >180 mg/dL (violet color). The dysglycemic rate for all patients (cumulated) is shown as the percentage of patients with at least one blood glucose measurement out of the glucose target range relative to the total number of patients in the respective period. The evolution of the relative patient number over time is plotted as a percentage of the total number on the right ordinate (**b**) in gray. The TWAG is shown as median ± 25th to 75th percentiles (box) with Tukey whiskers, data outside the Tukey range are shown as separate dots; mean is shown as dot within the Tukey box; n.s., not significant. The overall TWAG median glucose value pre-DCI (128.7 mg/dL) is shown as a red horizontal line for orientation in (**a**).

**Table 1 brainsci-14-00849-t001:** Patient characteristics and key facts about aSAH and ICU stay. Data are presented as mean ± SD or absolute numbers (with the percentage of the group [%]). In addition to the total study population, a split was made based on the occurrence of DCI. Significant *p*-values are depicted in bold.

Study Population		Delayed Cerebral Ischemia		
Variable	Total (*n* = 151)	No (*n* = 81)	Yes (*n* = 70)	*p*-Value
Demography				
Age [years]	55.2 ± 12.4	57.2 ± 12.6	52.9 ± 11.8	**0.035**
Sex (female)	101 (66.9%)	53 (65.4%)	48 (68.6%)	0.731
Body mass index [kg/m^2^] ^a^	26.3 ± 5.5	26.0 ± 4.9	26.5 ± 6.1	0.691
Medical history				
Diabetes mellitus type 2	7 (4.6%)	6 (7.4%)	1 (1.4%)	0.123
Hypertension	66 (43.7%)	39 (48.1%)	27 (38.6%)	0.254
Stroke history	7 (4.6%)	3 (3.7%)	4 (5.7%)	0.705
History of SAH	6 (4.0%)	3 (3.7%)	3 (4.3%)	1.000
Chronic kidney failure	1 (0.7%)	1 (1.2%)	0 (0.0%)	1.000
Chronic liver disease	3 (2.0%)	2 (2.5%)	1 (1.4%)	1.000
Psychiatric disorder	14 (9.3%)	9 (11.1%)	5 (7.1%)	0.575
Alcohol addiction	13 (8.6%)	6 (7.4%)	7 (10.0%)	0.578
Smoking history	51 (33.8%)	26 (32.1%)	25 (35.7%)	0.731
Sah scores				
Hunt and Hess score				
1–3	105 (69.5%)	60 (74.1%)	45 (64.3%)	0.293
4–5	46 (30.4%)	21 (25.9%)	25 (35.7%)	0.217
mFisher score				
1–2	72 (47.7%)	51 (63.0%)	21 (30.0%)	**<0.001**
3–4	79 (52.3%)	30 (37.0%)	49 (70.0%)	**<0.001**
Sah treatment				
Endovascular treatment ^b^	90 (59.6%)	49 (60.4%)	41 (58.6%)	1.000
Surgical clipping ^b^	57 (37.7%)	29 (35.8%)	28 (40.0%)	0.617
Time from onset to ICU admission [h]	10.1 ± 12.7	10.2 ± 13.9	9.9 ± 11.2	0.902
Ruptured aneurysm location				
Internal carotid	18 (11.9%)	7 (8.6%)	11 (15.7%)	0.213
Anterior cerebral	15 (9.9%)	8 (9.9%)	7 (10.0%)	1.000
Anterior communicating	52 (34.4%)	26 (32.1%)	26 (37.1%)	0.607
Middle cerebral	34 (22.5%)	19 (23.5%)	15 (21.4%)	0.846
Posterior communicating	15 (9.9%)	9 (11.1%)	6 (8.6%)	0.786
Vertebral	6 (4.0%)	4 (4.9%)	2 (2.9%)	0.686
Basilar	10 (6.6%)	7 (8.6%)	3 (4.3%)	0.341
Other	1 (0.7%)	1 (1.2%)	0 (0.0%)	1.000
ICU key facts				
SAPS II Score ^a^	37.4 ± 9.1	36.0 ± 9.0	39.1 ± 9.2	**0.039**
ICU length of stay [d]	25.2 ± 15.0	19.6 ± 10.8	31.7 ± 16.5	**<0.001**
ICU mortality	28 (18.5%)	9 (11.1%)	19 (27.1%)	**0.020**
Hospital length of stay [d]	30.2 ± 16.2	25.2 ± 12.1	36.1 ± 18.3	**<0.001**
Mechanical ventilation [h]	366.4 ± 417.5	235.8 ± 328.0	517.6 ± 459.3	**<0.001**

^a^ One missing value. ^b^ Two patients received no and two received a combination of endovascular and surgical treatment. mFisher score: Modified Fisher score, SAPS II: Simplified acute physiology score.

## Data Availability

The data presented in this study are available upon reasonable request from the corresponding author due to legal and ethical restrictions.

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
