# Peer review of "Value of Glycemic Indices for Delayed Cerebral Ischemia after Aneurysmal Subarachnoid Hemorrhage: A Retrospective Single-Center Study"

_brainsci, 2024, doi:10.3390/brainsci14090849_

Round 1

Reviewer 1 Report

Comments and Suggestions for Authors

In this retrospective study the authors present sound data on blood glucose levels in patients after aSAH comparing patients with DCI compared to those without. In contrast to numerous studies investigating in a suspected relation between these two variables the current study uses adequate and clear measures to depict the time course of blood glucose levels. They found significant differences in blood glucose levels between the two groups as well as before and after DCI. Although, as the authors themselves are mentioning, no causal dependence can be derived from these data, their findings may contribute to a better understanding of the pathophysiological mechanisms of DCI in the future. Hence, this study represents an important contribution in the field of DCI management. The manuscript is well written and the graphs are depicting the data clearly. 

Author Response

Thank you very much for your positive review of our manuscript. Please see the attachment for a point-by-point answer.

Reviewer 2 Report

Comments and Suggestions for Authors

The study involves a well-defined cohort of patients with aneurysmal subarachnoid hemorrhage (aSAH) in the potential role of glycemic indices in the timely detection of delayed cerebral ischemia (DCI) for prognosis. The identification of significant differences in glycemic indices between DCI and no-DCI patients is a notable contribution. I have several comments for authors to consider:

1. The retrospective design limits the ability to control for potential confounding variables. Additionally, the single-center focus may impact the generalizability of the findings. This should be stated clearly in the abstract and conclusion.

2. The study does not account for all potential confounders, such as medication use and nutritional status, that could influence the results. I suggest using propensity score matching to control for confounding variables further. 

3. The background section could benefit from a more detailed discussion of the existing literature and the specific gaps this study aims to address. Expand the literature review to provide a more comprehensive context for the study (DCI in the context of acute cerebrovascular disorders could also be briefly discussed (see PMID: 34553337). Clearly articulate the specific gaps in the current research that this study addresses.

4. The discussion lacks depth in addressing the limitations of the study and potential biases that may have affected the results. Provide a more detailed discussion of the study’s limitations and potential biases. Moreover, please discuss the broader implications of the findings for clinical practice and future research in more detail.

Minor comments:

6. Some sections, particularly the methods and results, could benefit from more concise writing to improve readability.

7. There are occasional inconsistencies in the presentation of statistical results (e.g., p-values) and the use of abbreviations. Ensure consistent formatting throughout the manuscript, particularly for statistical results and abbreviations. Standardize the presentation of p-values (e.g., always using two decimal places).

8. Some figures (e.g., TUDR graphs) may be unfamiliar to readers and require additional explanation. Provide a brief explanation of the less common figures in the figure legends or within the text to ensure they are easily understood by all readers.

9. There are a few self-citations. These citations appear relevant to the context of the study. Please review the self-citations to ensure they are necessary. 

Comments on the Quality of English Language

The manuscript contains formatting inconsistencies and could benefit from more concise writing in certain sections. Ensuring consistent formatting and improving clarity will enhance readability.

Author Response

Thank you very much for your detailled review of our manuscript. Please see the attachment for our point-by-point answer.

Reviewer 3 Report

Comments and Suggestions for Authors

The authors reported a retrospective, single-center, study about the Value of glycemic for delayed cerebral ischemia after aneurysmal subarachnoid hemorrhage.

after careful review, I have some suggestions:

- English needs minor revision (grammar and syntax);

- any abbreviations form needs a full form (or abbreviation list);

- the aim of the study should be clarified in the section Abstract;

- several indexes were recently explored as predictors of DCI or outcome in aSAH (triglyceride glucose index and others). The authors should mention this index and detail the pros and cons.

- A small sample size and other limitations should be described in a specific section. 

Comments on the Quality of English Language

mino revision of syntax

Author Response

Thank you very much for your detailed review of our manuscript. Please see the attachment for our point-by-point answer.

Round 2

Reviewer 2 Report

Comments and Suggestions for Authors

Thank you for addressing the points raised and incorporating the suggestions. No further comments. I recommend the manuscript be accepted for publication.